# A Case-Control Study of Hip Fracture Surgery Timing and Mortality at an Academic Hospital: Day Surgery May Be Safer than Night Surgery

**DOI:** 10.3390/jcm10163538

**Published:** 2021-08-12

**Authors:** Alim F. Ramji, Maxwell T. Trudeau, Michael R. Mancini, Matthew R. LeVasseur, Adam D. Lindsay, Augustus D. Mazzocca

**Affiliations:** Department of Orthopaedic Surgery, University of Connecticut Health Center, Farmington, CT 06030, USA; ramji@uchc.edu (A.F.R.); Michael.Mancini@quinnipiac.edu (M.R.M.); mlevasseur@uchc.edu (M.R.L.); alindsay@uchc.edu (A.D.L.); mazzocca@uchc.edu (A.D.M.)

**Keywords:** hip fracture, surgery, operative start time, day, night, mortality, case-control study

## Abstract

Time from hospital admission to operative intervention has been consistently demonstrated to have a significant impact on mortality. Nonetheless, the relationship between operative start time (day versus night) and associated mortality has not been thoroughly investigated. Methods: All patients who underwent hip fracture surgery at a single academic institution were retrospectively analyzed. Operative start times were dichotomized: (1) day operation—7 a.m. to 4 p.m.; (2) night operation—4 p.m. to 7 a.m. Outcomes between the two groups were evaluated. Results: Overall, 170 patients were included in this study. The average admission to operating room (OR) time was 26.0 ± 18.0 h, and 71.2% of cases were performed as a day operation. The overall 90-day mortality rate was 7.1% and was significantly higher for night operations (18.4% vs. 2.5%; *p* = 0.001). Following multivariable logistic regression analysis, only night operations were independently associated with 90-day mortality (aOR 8.91, 95% confidence interval 2.19–33.22; *p* = 0.002). Moreover, these patients were significantly more likely to return to the hospital within 50 days (34.7% vs. 19.0%; *p* = 0.029) and experience mortality prior to discharge (8.2% vs. 0.8%; *p* = 0.025). Notably, admission to OR time was not associated with in-hospital mortality (29.22 vs. 25.90 h; *p* = 0.685). Hip fracture surgery during daytime operative hours may minimize mortalities.

## 1. Introduction

Following hip fracture surgery, the 90-day mortality rate has been reported as 10–20% [1,2,3,4] with an in-hospital mortality rate of 1–5% [5,6,7,8]. The postoperative course is also highly morbid. A recent National Surgical Quality Improvement Program (NSQIP) study evaluating 31,738 hip fracture surgeries reported the median time in days from operative intervention to the occurrence of the most frequent complications, including myocardial infarction (2 days), stroke (3 days), pneumonia (4 days), pulmonary embolism (5 days), urinary tract infection (8 days), deep venous thrombosis (9 days), sepsis (11 days), superficial site infection (SSI) (16 days), deep SSI (23 days), and death (12 days) [9]. Additional postoperative complications include renal failure, hematoma, gastrointestinal hemorrhage, cerebrovascular accidents, arrhythmias, hemodynamic compromise, and infection requiring subsequent hardware removal [10,11,12,13].

Time from hospital admission to operating room (OR) has consistently been demonstrated to have a significant impact on mortality rates and postoperative outcomes. Decreased mortality rates have been observed in patients receiving surgery within 12 [14,15], 24 [6,12], 36 [16], 48 [17,18,19], and 72 [20,21,22] hours of hospital admission. Contrarily, operative delay to medically optimize patients may be essential and may not contribute to increased mortality [23,24,25]. Furthermore, traditional thinking regarding admission to OR time has also been reassessed in the setting of delaying procedures due to patients’ anticoagulation medication [26]. It is also believed that comorbid status may mediate the relationship between time to surgery and mortality [16,27]. Furthermore, other esoteric factors such as those involving the resources of the patient’s health care system and surgeon variability may also influence this relationship but are not fully captured by current research. Regardless, almost all of these studies did not account for operative start time (day versus night) in their analyses. The independent effect of operative start time on mortality has only been investigated in a handful of studies [28,29,30]. However, in the field of general surgery, a recent systematic review and meta-analysis analyzing ~3 million patients revealed that night surgery may be associated with increased mortality rates (adjusted odds ratio (aOR) = 1.47) [31]. Similar results were also seen in a multicenter retrospective cohort study of adult patients with night surgery demonstrating an increased risk of mortality (aOR = 1.26) [32].

The purpose of this study was to investigate the relationship between operative start time (day versus night) and associated mortality rates by controlling for time from admission to OR. Secondary objectives were to assess additional risk factors associated with 90-day mortality in this group. It was hypothesized that performing hip fracture surgery at night would be associated with an increased 90-day mortality.

## 2. Materials and Methods

### 2.1. Study Design

This study was a retrospective case-control study. The institutional review board at the University of Connecticut granted approval prior to initiation of the study. Data from April 2018 to July 2020 were retrospectively accrued, comprising a single institution’s experience with patients presenting with subtrochanteric, intertrochanteric, or femoral neck hip fractures. Inclusion criteria included patients 65 years of age or greater and isolated hip fractures resulting from low energy mechanisms of injury. Exclusion criteria included motor vehicle accidents or penetrating trauma, periprosthetic or bilateral hip fractures, and pathologic fractures resulting from primary or metastatic bone tumors. Operative fixation constructs included cephalomedullary nails, sliding hip screws, percutaneous hip screws, hip hemiarthroplasties, and total hip arthroplasties.

All procedures were performed at an academic hospital. For the entirety of the study, the attending surgeon was dictated by who was on call. Being a teaching hospital, the surgeries were primarily completed by a chief and junior resident, and this was the case for both day and night surgeries. Our institution has a clinical care pathway for fragility fracture patients with the goal of surgery within 48 h. Some of the major preoperative aspects of the protocol include limiting prolonged NPO (NPO reduced to only 6 h preceding surgery) and preoperative INR below 1.5. Postoperative aspects of the protocol include DVT prophylaxis beginning within 12–24 h, weight bearing and ambulation on the first postoperative day, and establishing physical therapy and orthopedic follow-up prior to discharge.

Clinical research associates under the supervision of the senior author collected preoperative patient comorbidities, operative details (type of anesthesia, fixation device, etc.), and 90-day postoperative outcomes/complications and mortalities. Time from admission to the operating room (OR) was calculated in hours. Patients were dichotomized into two independent study groups based on time of incision: (1) day operation—7 a.m. to 4 p.m.; (2) night operation—4 p.m. to 7 a.m. These time intervals correlate with the typical work hours for the OR staff at this institution. Operative time was calculated in minutes from procedure start to procedure closure. Intraoperative blood loss was estimated in milliliters by the attending surgeon at the conclusion of the case after consultation with the staff anesthesiologist.

### 2.2. Statistical Analysis

Descriptive statistics encompassing categorical variables are presented as frequencies and percentages, while continuous variables are presented as means and standard deviations. Pearson’s χ^2^ test or Fisher’s exact test and the Wilcoxon rank-sum test were used for categorical variables and nonparametric continuous variables, respectively. All tests were 2-sided. Clinical characteristics, operative details, and postoperative outcomes were compared between the two groups. Multivariable logistic regression modeling with backward stepwise elimination (*p* ≤ 0.10 for entry, *p* ≤ 0.10 for removal) was used to evaluate risk factors for 90-day mortality following hip fracture surgery. Analyses were performed with SPSS 25.0 (IBM Corporation, Armonk, NY, USA).

## 3. Results

### 3.1. Clinical Characteristics and Operative Details

From April 2018 to July 2020, 187 patients underwent hip fracture surgery at our institution. Of these, 14 were excluded for age, 2 were excluded for periprosthetic fractures, and 1 was excluded for mechanism of injury. The final population comprised 170 patients. These procedures were completed by 17 surgeons and 26 anesthesiologists, determined by who was on call. The mean age was 83.8 ± 8.08 years, and the mean body mass index was 24.97 ± 5.63 kg/m^2^. The majority of patients were female (69.4%) and ASA class 3 or 4 (71.2%). In terms of operative details, the average time from admission to OR was 26.0 ± 18.0 h; 71.2% of cases were performed between the hours of 7 a.m. and 4 p.m., comprising the day operation group, and 29.8% of patients were in the night group (Figure 1). Clinical characteristics for the included patients are listed in Table 1, Operative procedures included cephalomedullary nails (*n* = 73; 42.9%), cemented hip hemiarthroplasties (*n* = 68; 40.0%), percutaneous hip screws (*n* = 18; 10.6%), sliding hip screws (*n* = 8; 4.7%), and total hip arthroplasties (*n* = 3; 1.8%). Most importantly, there were no significant differences in preoperative comorbidities between the two operative groups. When controlling for the specific procedure, there were no significant differences in completion time (minutes) and intraoperative blood loss (mL) between day and night operations (Table 2).

### 3.2. Factors Associated with 90-Day Mortality

The 90-day mortality rate was 7.1% and the in-hospital mortality rate was 2.9%. Overall, night operations were associated with a significantly higher 90-day mortality rate (18.4% vs. 2.5%; *p* = 0.001) and accounted for 75% of all 90-day mortalities. Detailed operative start times associated with 90-day mortality are displayed in Figure 1. The factors significantly associated with 90-day mortality (*n* = 12) in univariable analysis are detailed in Table 3 and included age ≥85 years, male gender, night operation (OR start time between 4 p.m. and 7 a.m.), time from admission to OR, and congestive heart failure. Following the multivariable logistic regression, the only factor independently associated with increased risk for 90-day mortality was a night operation (aOR 8.91; *p* = 0.002; Table 3).

Furthermore, patients undergoing a night operation were significantly more likely to return to the hospital within 50 days of discharge (34.7% vs. 19.0%; *p* = 0.029) and experience mortality prior to discharge (8.2% vs. 0.8%; *p* = 0.025). These outcome metrics are summarized in Table 4.

### 3.3. Detailing of Mortalities Prior to Discharge from Night Operations

Most importantly, patients undergoing a night operation were 10 times more likely to experience mortality prior to discharge (*n* = 4; 8.2%). Notably, time from admission to OR was not significantly associated with in-hospital mortality (29.22 vs. 25.90 h; *p* = 0.685). Of the four patients that died, three (75%) had cardiac comorbidities (atrial fibrillation, cardiomyopathy, valvular disease, arrhythmias). Ultimately, the mean time from surgery to death was 1.75 days, with 3 of 4 (75%) of the mortalities resulting from cardiac arrest and one resulting from a stroke.

## 4. Discussion

The main finding of this study was that geriatric hip fracture surgeries performed at night had significantly higher rates of 90-day and in-hospital mortality than those performed during the day. Notably, operative start time appeared to have a stronger association with mortality in this series than time from admission to OR. In-hospital mortality following night operations (8.2%) was predominated by cardiac arrests occurring shortly after surgery.

In this series, the 90-day mortality rate was 7.1%, which is on par with current practice [2]. Overall, 29.8% of cases were performed as night operations. This is primarily a result of two factors: (1) our institutional protocol having a goal of surgery within 48 h and (2) increased OR availability beginning at 4:30 p.m. for hip fracture patients as mandated by our protocol. The factors associated with 90-day mortality in univariable analysis in this study included age ≥85 years, congestive heart failure, male gender, night operation (OR start time between 4 p.m. and 7 a.m.), and time from admission to OR. Expectedly, age was associated with an increased risk for mortality likely due to a decreased reserve capacity of the patient necessary to cope with the double hit of the hip fracture trauma and subsequent surgery [6]. Congestive heart failure was also identified as a potential risk factor, which is in accordance with previously published mortality analyses [1]. Lastly, time from admission to OR and night operations were also associated with increased risk for mortality. After controlling for all these factors simultaneously, the multivariable logistic regression revealed the only factor independently associated with 90-day mortality was a night operation. This is an interesting finding considering that most studies identifying time from admission to OR as a significant factor for mortality did not control for operative start time (day versus night) in their analyses [6,12,14,15,16,17].

Focusing on night operations (Table 4), it was also evident that patients undergoing night operations were almost twice as likely to return to the hospital within 50 days (*p* = 0.035). Night operations were significantly associated with an increased in-hospital mortality rate (8.4% vs. 0.8%; *p* = 0.029). Endo et al. previously identified the following factors to be associated with in-hospital mortality: age, timing of surgery, male sex, congestive heart failure, pulmonary circulation disease, renal failure, weight loss, and fluid and electrolyte disorders [5]. Conversely, time from admission to OR was not associated with in-hospital mortality in this study (29.22 vs. 25.90 h; *p* = 0.685).

These findings may be especially important to hospitals graded as level III–V trauma centers. It appears that operative performance is unaffected by starting during the day or at night, considering that there were no significant differences in implant usage, procedure length (minutes), and intraoperative blood loss. However, perioperative care involves a multitude of other factors that may vary between day and night operations for hip fracture patients that are not fully captured by this database but may play a larger role in mortality.

Possible underlying mechanisms that may be driving the association between night surgery and mortality include the following: extended NPO from waiting for surgery, unclear timing leading to suboptimal DVT prophylaxis, night surgery limiting postoperative weight bearing and ambulation, decreased staffing during night hours, and limited patient reserve capacity compounded by an altered circadian rhythm leading to poorer outcomes. However, it is hard to reconcile these notions as all patients followed an institutional hip fracture protocol irrespective of their operative start time (day versus night).

There are several noteworthy limitations to this study. First and foremost, this study consisted of a limited study population size resulting in the multivariable logistic regression identifying factors associated with 90-day mortality which provided a large 95% confidence interval (2.19–33.22) for night surgery, indicating some degree of imprecision. However, focusing on the absolute lowest end of the confidence interval, an adjusted odds ratio of 2.19 was still identified, warranting further investigation of this relationship. Furthermore, recent research has demonstrated ASA misclassifications, specifically underclassifications, can introduce bias, which in this study could be an additional factor increasing the adjusted odds ratio associated with night procedures [33]. While comorbidities were compared between the two groups, this study cannot account for possible confounders dictating clinical decision-making for selection and indication for day or night surgeries. This study additionally consisted of a retrospective design, which may have introduced confounding variables (e.g., nonuniformity of data collection) not typically seen in those created by a prospective study design. Moreover, some complications and possibly mortalities may have been missed if the patient presented to an alternative hospital or clinic. It is also acknowledged that patient level of mobility prior to hip fracture would be an additional variable worth controlling for; however, due to data collection limitations, this could not be considered.

## 5. Conclusions

These findings have important implications for the management of hip fracture patients and optimal timing of surgery. While delaying surgery may be associated with worse perioperative outcomes, the risk associated with operating at night for certain patients may be greater. Scheduling patients with complex comorbidities for day operations may better ensure optimal perioperative care and decrease in-hospital mortality rates. Future research is warranted to investigate this relationship further in order to better determine optimal hip fracture surgery timing and enhance our ability to care for these patients to decrease associated morbidity and mortality. Moreover, future research is needed to determine if these findings are present at other institutions and identify the underlying mechanisms governing the relationship between night surgery and mortality.

## Figures and Tables

**Figure 1 jcm-10-03538-f001:**
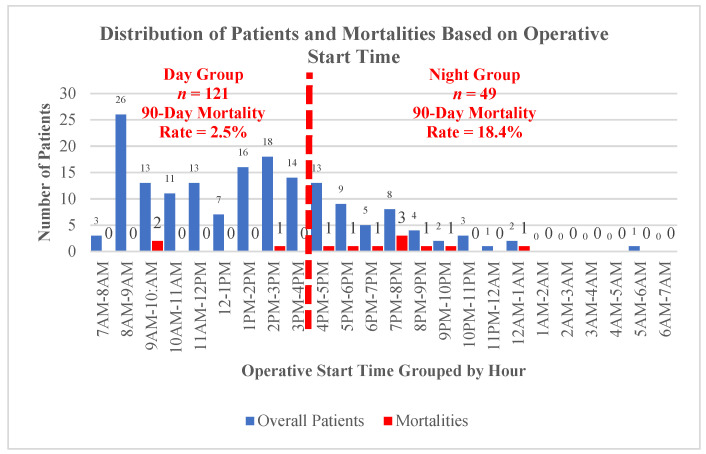
Distribution of patients and mortalities based on operative start time. Day operation group (7 a.m. to 4 p.m.) consisted of 121 patients with a 2.5% 90-day mortality rate. Night operation group (4 p.m. to 7 a.m.) consisted of 49 patients with an 18.4% 90-day mortality rate.

**Table 1 jcm-10-03538-t001:** Clinical Characteristics of Overall Study Group and Operative Start Time Groups (*n* = 170).

	Overall		Operative Start Time				
			Day Operation *n* = 121		Night Operation *n* = 49		
Variables	*n*	%	*n*	%	*n*	%	*p*
Age ≥ 85 years	84	(49.4)	59	(48.8)	25	(51.0)	0.789
BMI (kg/m^3^), mean (SD)	25.0	(5.6)	24.9	(5.9)	25.2	(4.9)	0.717
ASA, 3 and 4	121	(71.2)	82	(67.8)	39	(79.6)	0.123
Gender, female	118	(69.4)	88	(72.7)	30	(61.2)	0.140
Type of fracture							0.738
Femoral neck	91	(53.5)	67	(55.4)	24	(49.0)	
Intertrochanteric	72	(42.4)	49	(40.5)	23	(46.9)	
Subtrochanteric	7	(4.1)	5	(4.1)	2	(4.1)	
Type of procedure							0.986
Cephalomedullary nails	73	(42.9)	52	(43.0)	21	(42.9)	
Sliding hip screws	8	(4.7)	5	(4.1)	3	(6.1)	
Hemiarthroplasty	68	(40.0)	49	(40.5)	19	(38.8)	
Percutaneous hip screws	18	(10.6)	13	(10.7)	5	(10.2)	
Total arthroplasty	3	(1.8)	2	(1.7)	1	(2.0)	
Type of anesthesia							0.791
General	117	(68.8)	84	(69.4)	33	(67.3)	
Spinal	53	(31.2)	37	(30.6)	16	(32.7)	
Admission to OR time (hours), mean (SD)	26.0	(18.0)	25.5	(15.2)	27.1	(23.6)	0.603
Procedure time (minutes), mean (SD)	96.0	(43.6)	97.2	(43.7)	92.9	(43.8)	0.564
Blood loss (mL), mean (SD)	175.9	(122.0)	168.6	(112.6)	192.2	(140.9)	0.282
Asthma	8	(4.7)	8	(6.6)	0	(.0)	0.107
Atrial fibrillation	52	(30.6)	32	(26.4)	20	(40.8)	0.066
CHF	23	(13.5)	14	(11.6)	9	(18.4)	0.321
COPD	17	(10.0)	11	(9.1)	6	(12.2)	0.576
CAD	18	(10.6)	14	(11.6)	4	(8.2)	0.594
Diabetes	20	(11.8)	13	(10.7)	7	(14.3)	0.600
Delerium	18	(10.6)	13	(10.7)	5	(10.2)	0.917
Dementia	45	(26.5)	35	(28.9)	10	(20.4)	0.254
GERD	22	(12.9)	15	(12.4)	7	(14.3)	0.802
HTN	105	(61.8)	71	(58.7)	34	(69.4)	0.193
History of cancer	23	(13.5)	15	(12.4)	8	(16.3)	0.621
Hypothyroidism	31	(18.2)	23	(19.0)	8	(16.3)	0.827
Osteoarthritis	13	(7.6)	7	(5.8)	6	(12.2)	0.201
Osteoporosis	27	(15.9)	21	(17.4)	6	(12.2)	0.409
Parkinson’s	8	(4.7)	5	(4.1)	3	(6.1)	0.691

Abbreviations: ASA, American Society of Anesthesiologists; BMI, body mass index; CAD, coronary artery disease; CHF, congestive heart failure; COPD, chronic obstructive pulmonary disease; GERD; gastroesophageal reflux disease; HLD, hyperlipidemia; HTN, hypertension; OR, operating room; SD, standard deviation.

**Table 2 jcm-10-03538-t002:** Procedure Time and Intraoperative Blood Loss Comparison for Day and Night Operation Groups (*n* = 170).

	Operative Start Time				
	Day Operation *n* = 121		Night Operation *n* = 49		
Variables	Mean	SD	Mean	SD	*p*
Type of procedure					
Cephalomedullary nails (*n* = 73)					
Procedure time (minutes)	83	(36.1)	78	(38.4)	0.572
Blood loss (mL)	149	(86.2)	158	(89.1)	0.716
Sliding hip screws (*n* = 8)					
Procedure time (minutes)	51	(13.5)	66	(26.5)	0.323
Blood loss (mL)	55	(38.9)	100	(50.0)	0.235
Hemiarthroplasty (*n* = 68)					
Procedure time (minutes)	124	(32.7)	113	(38.5)	0.243
Blood loss (mL)	222	(116.9)	232	(151.1)	0.785
Percutaneous hip screws (*n* = 18)					
Procedure time (minutes)	52	(12.8)	73	(34.7)	0.071
Blood loss (mL)	55	(26.5)	83	(28.2)	0.132
Total arthroplasty (*n* = 3)					
Procedure time (minutes)	212	(17.0)	207	NA	0.850
Blood loss (mL)	325	(176.8)	650	NA	0.374

**Table 3 jcm-10-03538-t003:** Univariable and Multivariable Analysis for 90-Day Mortality.

	Univariable Analysis							Multivariable Analysis
	No		Yes					
Variables	*n*	%	*n*	%	*p*	Adjusted Odds Ratio	(95% CI)	*p*
Age					0.078			
<85 years	83	(96.5)	3	(3.5)		REF		
≥85 years	75	(89.3)	9	(10.7)		3.43	(0.82–14.31)	0.091
Gender					0.047			
Female	113	(95.8)	5	(4.2)		REF		
Male	45	(86.5)	7	(13.5)		2.24	(0.59–8.47)	0.234
CHF					0.060			
No	139	(94.6)	8	(5.4)		REF		
Yes	19	(82.6)	4	(17.4)		2.01	(4.51–8.98)	
Admission to OR time (hours), mean (SD)	24.9	(15.5)	40.2	(35.7)	0.004	1.03	(0.99–1.06)	0.061
Operative start time					0.001			
Day operation	118	(97.5)	3	(2.5)		REF		
Night operation	40	(81.6)	9	(18.4)		**8.91**	**(2.19–33.22)**	**0.002**

All variables from Table 1 were included in this analysis, but those not displayed did not obtain statistical significance required for entry. Abbreviations: CHF, congestive heart failure; CI, confidence interval; NA, not applicable; OR, operating room; SD, standard deviation. Bolding indicates significance.

**Table 4 jcm-10-03538-t004:** Comparison of Various Outcome Metrics for Day and Night Operations (*n* = 170).

	Day Operation *n* = 121		Night Operation *n* = 49		
Outcome Metrics	*n*	%	*n*	%	*p*
Any complication prior to discharge	33	(27.3)	17	(34.7)	0.336
Return in 90 days	39	(32.2)	20	(40.8)	0.287
Return in 50 days	23	(19.0)	17	(34.7)	**0.029**
Relevant readmission	7	(5.8)	3	(6.1)	1.000
Mortality, 90 days	3	(2.5)	9	(18.4)	**0.001**
In-hospital mortality	1	(.8)	4	(8.2)	**0.025**
Transfusion postoperatively	18	(14.9)	6	(12.2)	0.809
LOS (days), mean (SD)	3.5	(3.0)	3.5	(2.7)	0.995

Abbreviations: LOS, length of stay; OR, operating room; SD, standard deviation. Bolding indicates significance.

## Data Availability

The data presented in this study are available on request from the corresponding author. The data are not publicly available due to ethical reasons.

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
