# Peer review of "A Case-Control Study of Hip Fracture Surgery Timing and Mortality at an Academic Hospital: Day Surgery May Be Safer than Night Surgery"

_jcm, 2021, doi:10.3390/jcm10163538_

Round 1

Reviewer 1 Report

My browser does not support pasting the comments. My comments are attached in the Word file. Thank you 

Author Response

Comment: I prefer adding the study design and setting in the title and keywords. 

Response: Thank you for this suggestion, we believe it would be an improvement to add the study design to the title and keywords to make it more evident. Therefore, we have adjusted the title to now read, “Hip Fracture Surgery Timing and Mortality At An Academic Hospital: Day Surgery May Be Safer Than Night Surgery Based on a Case Control Study.” (lines 1-4). Furthermore, we have added case control study to the keywords list. 

Comment: Introduction is well written. Kindly elaborate on the rationale behind conducting this study. Why was the study conducted? I am sure there would be multiple factors, kindly elaborate.

Response: Thank you for this suggesiton. Accordingly, we have added the following statement to the introduciton, “However, in the field of general surgery, a recent systematic review and meta-analysis analyzing ~3 million patients revealed that night surgery may be associated with increased mortatliy rates (adjusted odds ratio [aOR]=1.47).[31] Similar results were also seen in a multicenter retrospective cohort study of adult patients with night surgery demonstrating an increased risk of mortality (aOR=1.26).[32]” (lines 52-56). 

Comment: Did you include all patients irrespective of their comorbids, ASA status or their Charlson Comorbidity Index? That can lead to misclassification bias. Try to elaborate on that.

Response: Thank you for this suggestion. We have added the following statement to the limitations section of the manuscript, “Furthermore, recent research has demonstrated ASA misclassificaitons, specifically, under-classifications, can introduce bias, which in this study could be an additional factor increasing the adjusted odds ratios associated with night procedures.[33]” (line 203-206). 

Comment: The only major flaw of the study is analysis method (logistic regression reporting OR). I do believe that wrong tests were used requiring the paper to be reviewed by a biostatistician.  Details are below: 
In the subheading 2.1 Study design, you mentioned everything but the study design. I had to jump to the results and tables to figure out whether it was a case control study or retrospective cohort with comparison group. Despite that, you are mentioning the word “cohort” implying it’s a cohort study with two groups based on the primary exposure of “Night/after duty surgeries”. If so, then why are you reporting the Odds Ratio in place of the Relative risk? 
If your study is case control study with 2 groups based on the outcome “mortality” then I agree with the analysis using logistic regression and reporting OR. 

Response: Thank you for this keen observation and comment. This study was in fact a case control study. To make this evident, we have altered section 2.1 Study Design to immediately state, “This study was a retrospective case control study.” (line 64). Therefore, we agree with you that logistic regression and reporting aOR was the appropriate method. 

Moreover, now after you raise this concern, we completely understand that using the terminology “cohort” is problematic and gives the reader the faulty impression that it is a cohort study. Therefore, we have altered the terminology to refer to day and night cohorts rather as “groups”, and have altered this as such throughout the manuscript. 

Comment: Can you give references why is it 8 hours for the NPO time? The usual accepted recommendation is 6 hours NPO. Who was the surgeon performing these procedures? Faculty or residents? What is average or minimum clinical experience? Kindly mention whether night procedures were done by the on-call residents or faculty himself? Usually, residents operate after duty hours and faculty operates at daytime. That can lead to different outcomes. It’s a limitation of almost all academic hospitals. However, I believe it needs to be mentioned in the methodology.

Response: Thank you for these comments. You are correct, the usual accepted recommendation is 6 hours NPO, furthermore our protocol is no longer than 6 hours NPO. This was actually an error and has been corrected to correctly state “NPO reduced to only 6 hours preceding surgery” (line 77). Thank you for this close reading. 

These procedures were completed by 17 surgeons, non of which were associated with an increased mortality rate (p=0.559). The attending surgeons, for the entirety of the study, was determined by who was on call. It should be noted that this is a teaching hospital so the surgeries were primarily completed by a chief and junior resident – this does not change whether it is a day or night operation. We agree that this important to share in the manuscript and have therefore altered the methods to state, “For the entirety of the study, the attending surgeon was dictated by who was on call. Being a teaching hospital, the surgeries were primarily completed by a chief and junior resident and this is the case for both day and night surgeries.” (lines 74-77). 

Comment: Again I stress on the point of utilizing proper tests to avoid wrong conclusions. The groups were based on primary exposure. That makes it a cohort study. Logistic regression is not the correct method analyzing the cohort design. Better to do Cox regression reporting RR. If your outcome is rare, then RR=OR. However, it’s not in your case, unless you justify that maybe. Moreover, only one variable is significant at your multivariable model. That makes it a univariable model, not multivariable unless you report the Adjusted OR (aOR) if other variables are included.

Response: Thank you again for these keen observations and comments. Again, it was a case control study, so we believe it is best to use the logisitc regression. 

Furthermore, there was only one variable that was significant in the multivariable model, and we reported adjusted odds ratio. However, we understand your point as this was not made evident in the manuscript as we solely referred to it as an”odds ratio”. To prevent further confusion, we have adjusted all terminology to state “adjusted odds ratio” and “aOR”. We understand your concern and believe this makes the results much easier to comprehend.

Reviewer 2 Report

This study evaluates hip fracture surgery timing and mortality in a tertiary hospital. The topic is interesting and actual. Time from hospital admission to surgery has an impact in mortality rates after hip fracture. However, optimize patient to surgery is also essential and it seems not to increase mortality rates. 

The results obtained are clearly presented and according to the methods. In my opinion there are others factors that have an influence in hip fracture mortality like mental state, level of mobility, early mobilisation, hospital re-admission. And maybe these factors influenced the rate of mortality between  day and night operation. Authors should include all factors related to hip mortality and compare them among both groups.

Author Response

This study evaluates hip fracture surgery timing and mortality in a tertiary hospital. The topic is interesting and actual. Time from hospital admission to surgery has an impact in mortality rates after hip fracture. However, optimize patient to surgery is also essential and it seems not to increase mortality rates. 

The results obtained are clearly presented and according to the methods. In my opinion there are others factors that have an influence in hip fracture mortality like mental state, level of mobility, early mobilisation, hospital re-admission. And maybe these factors influenced the rate of mortality between  day and night operation. Authors should include all factors related to hip mortality and compare them among both groups.

Response: Thank you for these comments and suggestions. We agree that there are other factors that may be impacting mortality rate amongst this study population. Regarding mental state, we have edited the manuscript to now include delirium on presentation (Table 1, page 4). We believe this is an important edition to the manuscript and appreciate the suggestion. However, there was no significant difference between the day and night groups (10.7% vs. 10.2%, p=0.917). Furthermore, delirium was not associated with 90-day mortality (p=0.619).

Level of mobility would also be an interesting variable to analyze, but unfortunately our database does not possess this information. Therefore, we have added the following statement to the limitations section, “It is also acknowledged that patient level of mobility prior to hip fracture would be an additional variable worth controlling for, however, due to data collection limitations this could not be considered.” (lines 206-208). Conversely, early mobilization is part of our standardized hip fracture protocol, which includes weight bearing and ambulation of the first postoperative day – this aspect of the protocol was described in our original submission (lines 70-76).

Lastly, hospital readmission rates were compared between the day and night cohorts (page 6, Table 4). Overall, there was no difference in return to hospital in 90 days between day and night surgeries (32.2% vs. 40.8%, p=0.287), but there was a significant difference seen in return to hospital in 50 days (19.0% vs. 34.7%, p=0.029). We do not believe that this necessarily is a contributor to mortality but rather an additional marker of morbidity associated with night surgery.

Round 2

Reviewer 1 Report

Thank you for responding to the comments. Just I recommend using "A case control study" than "based on case control study".

Author Response

Thank you for this suggestion. We have altered the title to "A Case-Control Study of Hip Fracture Surgery Timing and Mortality At An Academic Hospital: Day Surgery May Be Safer Than Night Surgery". We believe this reads much better and makes the methodology very explicit. 

Reviewer 2 Report

Thank you very much for your answer. It is a pity that you do not have recorded the level of mobility in hip fracture patients. I recommend you to start recording it and continue with your research in the influence of hip fracture timing in mortality. It helps to have more information about the best moment for the hip surgery.

Author Response

Thank you for this comment. We agree it is unfortunate that we have not routinely collected this data. It would surely aid this study and future hip fracture studies. To that point, we agree with your advice and plan to start recording it as we move forward.